# Sustainable Glass Recycling Culture-Based on Semi-Automatic Glass Bottle Cutter Prototype

**Jovheiry García Guerrero [1], Juvenal Rodríguez Reséndiz [1,2,\*], Hugo Rodríguez Reséndiz [1], José Manuel Álvarez-Alvarado [1] and Omar Rodríguez Abreo [2,3]**

1    Facultad de Ingeniería, Universidad Autónoma de Querétaro, Querétaro 76010, Mexico; jgarcia95@alumnos.uaq.mx (J.G.G.); hugorore@uaq.mx (H.R.R.); jmalvarez@uaq.mx (J.M.Á.-A.)
2    Red de Investigación OAC Optimización, Automatización y Control. El Marques, Queretaro 76240, Mexico; omar.rodriguez@upq.edu.mx
3    Industrial Technologies Division, Universidad Politécnica de Querétaro, Santiago de Querétaro 76240, Mexico
\*    Correspondence: juvenal@uaq.edu.mx

**Abstract:** Humanity has developed recycling activities over time due to their benefits, the shortage of raw materials, or the footprint with regard to the environment. The absence of a recycling culture in Mexico has not allowed its development and growth despite the benefits. In 2012, Mexico only recycled less than 10% of urban solid waste. Most recycling activities are focused on plastic, paper, and cardboard products due to their prices in local markets. This article presents a semi-automated prototype focused on recycling glass bottles using the thermal shock phenomenon. It aims to develop a sustainable glass recycling culture by creating a new branch for the integral glass recycling process and a proposal base on Integrated Sustainable Waste Management (ISWM) and the Quintuple Helix Model. It helps to reduce waste and resource recovery from recycling and upcycling glass bottles. The products obtained from upcycling fulfill new uses and acquire new value, while glass leftovers continue the integral recycling process for glass. Additionally, this paper demonstrates the relation between the ISWM and the Quintuple Helix Model and the opportunity to implement the twelfth Sustainable Development Goal (SDG).

**Keywords:** glass bottle upcycling; resource recovery; COVID-19; glass waste recycling; sustainable development

## 1. Introduction

Since the first permanent establishments of man, recycling has been conducted. They created tools from elements such as wood and stone. This type of behaviour was also exhibited during the pre-Hispanic era, with tools produced with bones of animals, rocks, and wood [1]. Even after the emergence of metallurgy, recycling waste and unusable metal objects were used. In China, around 2500 B.C., metal was recycled. A similar situation occurred in ancient Rome, where glass was recycled to make their famous mosaics [2]. The fact that the processing of raw materials and their transformation became quick and straightforward tasks overshadowed reusing and recycling.

Only in times of war did people take an interest in reusing materials and recycling recovered objects. During the War of Independence in North America, recycling became a frequent practice at the end of the 18th century. It was in the Second World War when official recycling campaigns were launched in several countries. After the war, these practices were forgotten [3].

One of the crucial global challenges in reducing the carbon footprint is the increasing demand for natural resources and landfill spaces in waste glass disposal activities [4]. Nowadays, the culture of glass recycling has been increasing worldwide because it has been committed to developing technologies to comply with environmental policies [5].

In this context, it is necessary to create new recycling glass strategies because it is not biodegradable and remains stable for a long time [6].

### 1.1. Glass Recycling Insight

The recycling of glass brings numerous benefits. These are introduced in the following points:

- Reduction in raw material extraction: cullet, glass crashed into small pieces through recycling process steps, serves as a substitute for the glass production process. This allows a reduction in raw material extraction [7].
- Save energy: Using cullet in the glass production process reduces energy consumption. It brings energy savings from avoided provision and calcination of raw materials [7]. Cullet has a lower melting point compared to its raw material [8]. When cullet is used in closed-loop cycles, generally, a cullet volume increase of 10% in the glass-making process reduces energy consumption by 2–3% [9]. The Foundation of the Energy of the Community of Madrid (FENERCOM, its acronym in Spanish) mentions that recycling glass saves 32% of energy in the process, and *ecovidrio* talks about a saving of 38% [10]. A case of cullet recycling in Italy exposed a volume reduction of 318 $Mm^3$ of natural glass.
- Quality conservation: Glass has the benefit of conserving its purity and undergoing no loss of quality when it is recycled [11]. The amount of new glass is the same as the amount of glass used for recycling.
- Waste reduction: Cullet brings a volume reduction in waste in landfills. A glass bottle exposed to the environment takes more than 4000 years to disappear. According to the National Association of Manufacturers of Glass Containers (ANFEVI, its acronym in Spanish), recycling a glass container that has finished its period of life did not increase the volume of Urban Solid Waste (USW) [12].
- Decrements in environmental contaminants: Glass recycling is one technique that positively impacts sustainability since it is possible to reuse this raw material several times. Turning it into an alternative raw material by not losing its quality, which makes it an attractive proposal [13]. According to [14,15], when a glass tone is recycled, 0.58 t of $CO_2$ is saved, 20% reduces air pollution, and water pollution is cut from 40 to 50% throughout the supply chain. *Ecovidrio* mentions savings of 53% of $CO_2$ emissions when using cullet versus raw materials. The Italian cullet recycling case proved a reduction of 1.9 Mt of $CO_2$ while, as a general rule, an increase of 10% of cullet volume decreases $CO_2$ by 5%. [7].

ANFEVI mentions that recycling glass is also known to be integral because it does not generate waste. New products do not lose their qualities and properties, offering multiple benefits (energy, resources, and waste reduction). Figure 1 adapts the integral glass recycling process according to ANFEVI.

Despite the benefits of glass recycling, the materials most in demand by the recycling industry are paper and plastic due to their environmental impact and other facts. This has caused the frequency of glass recycling to reduce.

### 1.2. The Case of Glass Recycling in Mexico

In the case of Mexico, glass recycling is an activity that is not very well rewarded compared with similar activities. According to the Ministry of Environment and Natural Resources of Mexico (SEMARNAT, its acronym in Spanish), approximately 42.1 kt of Urban Solid Waste (USW) was generated in 2012. This year, glass occupies the fourth position of the USW recollected in 2012 with 5.9% of the total [16]. Data from 2017 indicate a recollection of 44.6 kt. Additionally, glass occupies the same position but with 6.1% [17].

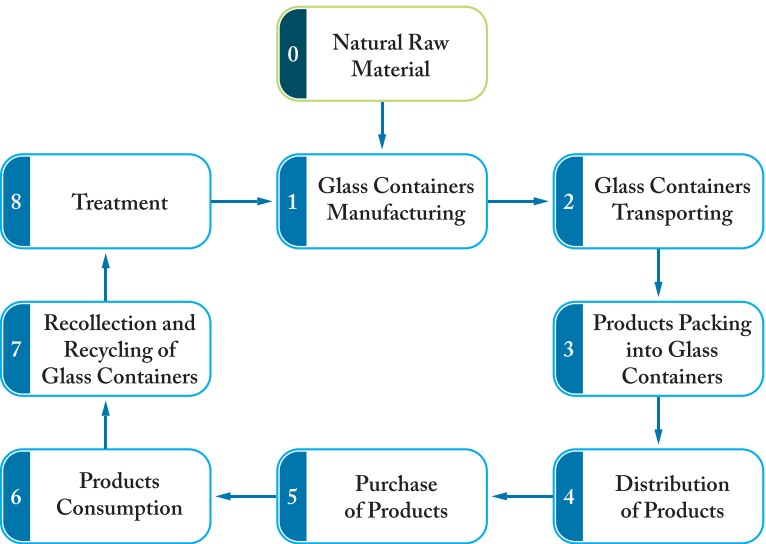

**Figure 1.** The integral glass recycling process.

In the case of the composition of valuable USW collected, glass occupied the fourth position (13.8% of the total in 2012) [16] and the third position with 18.5% in 2016 [17]. Appendix A.1, Table A1 exhibits the composition of the USW recollected in 2012 and 2017, and Table A2 presents the Valuable USW composition from 2012 and 2016.

According to data, there is an increase of 4.7% for glass, but in reality there is a reduction of 14,188 kg per day. Despite an increment in the amount of USW recollected, a reduction in the amount of valuable USW is notable. There are factors affecting participation levels in recycling programs. One of these is glass recycling programs, which is the price that scrap and waste of glass have in the local market. Table 1 contains the prices of both offered by *Supraciclaje* [18], a recycling company dedicated to the buy and sale of scrap.

**Table 1.** Prices for scrap and waste materials.

| Scrap | | Waste | |
| --- | --- | --- | --- |
| Material | USD/kg | Material | USD/kg |
| PET | 0.12–0.47 | Plastic | 0.12–0.54 |
| Plastics | 0.06–0.09 | Archival paper | 0.17 |
| Paperboard | 0.07 | Cardboard | 0.12–0.15 |
| Paper | 0.005–0.05 | Glass | 0.06 |
| Glass | 0.005 | | |

Prices are approximate. Currency exchange equivalent 20.43 MXN to 1 USD. Data consulted on 29 January 2021.

Organizations and individuals skip recollection of glass in recycling campaigns due to its low price. They think recycling glass is not worthy in comparison with similar activities. According to the *Cerrando el Ciclo* website, 2.5 Mt of glass bottles are generated annually, but only 12% of these are recycled [19]. In 2012, Mexico only recycled 9.6% of the USW collected [16]. Table 2 compiles the percentage of USW recycled in Mexico in the years 2012, 2014, 2016, and 2018 according to data by INEGI [20].

According to the Organisation for Economic Co-operation and Development (OECD), there are no data related to recycling percentage in Mexico for years 2012, 2014, 2016, and 2018 [21]. Table A3 presents the percentage of USW recycled in various countries from OECD.

Similarly, the growing need to accelerate efforts to meet global agreements such as the SDG ethically require solutions to be proposed [22]. Indeed, the use and development of technology with social responsibility creates a more significant impact not only on the application, but also on the environment.

**Table 2.** Percentage of USW recycled in Mexico.

| Year | USW Recollected t/day | USW Sent for Recovery t/day | USW Recovered t/day | Recollected/Sent for Recovery % | Sent for Recovery/ Recovered % | Recollected/ Recovered % |
|------|------|------|------|------|------|------|
| 2012 | 99,770,725.0 | 6,083,199.0 | 764,596.0 | 6.10 | 12.57 | 0.77 |
| 2014 | 102,887,315.0 | 3,423,613.0 | 510,786.0 | 3.33 | 14.92 | 0.50 |
| 2016 | 104,734,930.0 | 5,375,859.0 | 2,117,472.0 | 5.13 | 39.39 | 2.02 |
| 2018 | 107,055,547.0 | 7,339,967.0 | 1,948,168.0 | 6.86 | 26.54 | 1.82 |

In 2015, the 2030 Agenda for sustainable development was adopted by all United Nations Member States. Its scale and ambition are demonstrated by The 17 Sustainable Development Goals (SDGs) and their targets. Additionally, they balance the dimensions of sustainable development (environmental, social, and economic) [23]. The SDGs recognize that strategies must be used to end poverty and other deprivations by working to preserve our oceans and forests and tackle climate change. The strategies must reduce inequality, improve health and education, and encourage economic growth [24].

In the case of Mexico, Victor M. Toledo, former secretary of SEMARNAT, has visualized a set of actions already undertaken by the Mexican government to achieve at least ten of the seventeen goals of the 2030 Agenda [25]. Some targets from goals 1, 2, 3, 4, 5, 8, 9, 11, 16, and 17 are used in all 31 States and one Federal District. These and some more targets from goals 6, 7, 10, 13, 14, and 15 are used in the United States of Mexico [26]. The SDG number 12 is not used, although it aims to ensure sustainable production and consumption patterns.

*1.3. Understanding the Project*

The principal objective is to develop a sustainable glass recycling culture proposal using a semi-automatic glass bottle cutter prototype. The project focuses on individuals and small organizations to improve glass recycling, divulge its benefits, increase participation levels, and transform it into a Technological Pedagogical Content Knowledge (TPACK) [27]. It is used for upcycling, a term whose origin combines the words "upgrading" (to add value) and "recycling" (to reuse). Upcycling combines the recycling and reusing of objects and waste to convert them into new ones or artistic creations [28]. Additionally, it generates little glass waste because not all glass bottles are used for new objects. Therefore, the proposal needs to consider the glass waste generated by upcycling to reach zero waste production. The proposal is based on the Faculty of Engineering case using Integrated Sustainable Waste Management (ISWM) and the Quintuple Helix Model. Both offer a comprehensive opportunity for sustainability development. It proposes using glass bottles in perfect conditions for upcycling while glass waste is produced by upcycling and glass bottles are in a deteriorated state for cullet production. The prototype uses the thermal shock phenomenon for glass bottle upcycling.

This work is divided into the following sections: Section 2 contains essential information to understand the project. Section 3 introduces technical information for the development of the methodology. Section 4 presents and discusses the results. Section 5 establishes the conclusions of the paper.

**2. State of the Art**

Section 2.1 shows the ISWM and Section 2.2 introduces the Quintuple Helix Model. Both are frameworks used for the proposal development. Section 2.3 presents information about glass bottles, Section 2.4 explains the phenomenon used for glass upcycling, and Section 2.5 details technical information for the temperature control.

*2.1. Integrated Sustainable Waste Management*

According to the objective of zero glass waste reduction, it is necessary to consider all processes related to glass bottle upcycling and recycling. A comprehensive analysis of these involved in glass waste management allows us to design the proposal.

The ISWM framework is based on the hierarchy of waste management. It aims to reduce, reuse, recycle, and recover. This framework considers effectiveness, efficiency, equity, and sustainability as principles. According to [29], the Integrated Solid Waste Management framework becomes an ISWM when sustainability is required. The ISWM considers three dimensions: elements, stakeholders, and aspects. Table 3 details the components of each dimension.

**Table 3.** Dimensions and principles of ISWM framework.

| Dimensions | | |
|---|---|---|
| **Elements** | **Stakeholders** | **Aspects** |
| Generation and Separation | Local/Regulatory Authorities | Environmental |
| Recovery | Non-Governmental Organizations (NGOs) | Institutional |
| Collection | Community-Based Organizations (CBOs) | Political/Legal |
| Transfer | Service Users | Financial/Economic |
| Treatment and Disposal | Informal/Formal Sector | Socio-Cultural |
| Reduce, Recycle, and Reuse | Donor Agencies | Technical and Performance |

Each dimension needs to be addressed at the same time when the ISWM is designed.

### 2.2. The Quintuple Helix Model

By analyzing all aspects of the ISWM, the relation with the Quintuple Helix Model is highlighted. All stakeholders from ISWM are related to each helix from the Quintuple Helix Model. Additionally, five of the aspects represent the five helices.

The Quintuple Helix Model represents the embedded Quadruple Helix Model (QHM) and the Third Helix Model (THM) in the context of the environment. The focus of the THM is the relations between university, government, and industry, while the attention of the QHM adds to the media-based and culture-based public. Then, the five helices are education system, economic system, political system, media-based and culture-based public, and natural environment. The Quintuple Helix Model is considered as an interdisciplinary and transdisciplinary framework at the same time. It requires the continued involvement of the social and humanities sciences through to the natural sciences and a complete analytical understanding of the five helices. This model helps provide an analytical model for social ecology and sustainable development [30].

According to [31], in the context of global warming and promotion of sustainable development, feasible, sustainable development is described when investments flow in the education system helix. It creates suggestions and impulses for knowledge creation and impacts human capital for sustainable and greener development, which serves as new knowledge for the economic system. In the economic helix, new types of green services and products exist with the creation of new types of jobs.

The know-how and sustainability of the economic system serve to protect the environment. It means a reduction in resource exploitation, contaminants, and destruction. The new know-how obtained by preserving the natural environment provides a new, greener lifestyle and nature knowledge for the media-based and culture-based public. This helix requires spreading the new knowledge gained and implementing it in a conscious, affordable, and simple way. The political system uses the new information of citizens about their necessities, satisfaction, and problems. Additionally, it compiles the knowledge from the other three helices. The political helix aims to create new legal and political capital, which serves for new investments, suggestions, and objectives offering sustainability. This new knowledge and know-how flow back into the previous four helices. Thus knowledge circulates in the model.

### 2.3. Glass Bottle

It is necessary to understand the shape of glass bottles for the prototype design and its classification for separation and treatment processes due to glass bottles being the focus for glass recycling.

The first glass containers, focused on the conservation and storage of specific products, were developed by the Romans. They improved the blowing technique used by the Egyptians in around 200 B.C. It was in the mid-seventeenth century that Sir Kenelm Digby designed the modern glass bottle. The design consisted of a cylindrical body with fallen shoulders topped by a long neck. The three most commonly used types of glass containers are used for liquids, with a narrow neck of a diameter smaller than 35 mm on solid foods, jams, and compotes, with a diameter greater than 35 mm, and pharmaceutical, cosmetic, chemical, and perfume products [32]. Figure 2 shows the scheme of industrial glass bottles and jars.

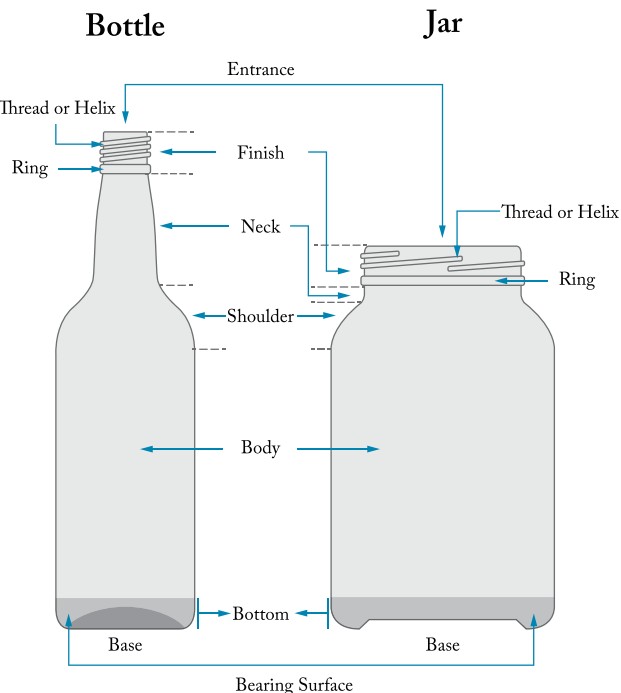

**Figure 2.** Industrial glass bottle and jar scheme.

Glass beverage containers are classified according to their color, mainly amber/brown, green, and crystalline [33]. The color of the glass is one of the criteria for classified glass in recycling plants. Other criteria are the presence of inorganic and non-magnetic foreign materials, magnetic metals, aluminum, lead, organic matter, labels, paint on glass, and contamination (garbage, moisture, sand, earth, ant, or lime) within the proposed limits [34].

### 2.4. Thermal Shock

Thermal shock is the phenomenon of the glass upcycling process. Understanding it is vital to determine how the prototype must work to produce thermal shock during the process.

Thermal shock is a phenomenon that generates great efforts in refractory material by temperature cycles. The effects produced generate cracks in the material. These vary depending on material working conditions, dimensions, geometry, and temperature cycle. There is no universal and straightforward test for studying its behavior and extrapolating to these factors [35]. Table 4 presents a description of these factors.

The temperature variation presented on the glass bottle is equal to the maximum temperature difference with the initial temperature on the glass bottle. It is known as temperature gradient. Additionally, the time that it takes the temperature gradient to occur is known as the time gradient.

The cracks generated are used for the glass upcycling process. These are occur when controlling the temperature of the glass bottle.

**Table 4.** Factors that affect the thermal shock phenomenon in refractory materials.

| Factors | Interpretation |
|---|---|
| Working conditions | Conditions in which the material is elaborated |
| Dimensions | Thickness of the refractory material |
| Geometry | The shape of the object |
| Temperature cycle | A cycle where temperature varies |

### 2.5. Temperature Control

The prototype must control the heating process to keep the temperature of the heating element at the same value during the process. The user does not have to pay attention to control the temperature, which causes a reduction in efforts of the user. It is the reason why automatic control is used to achieve this task.

Automatic control systems result from the interconnection of elements formed within a system in such a way that the arrangement can control itself. The system, or some of its components, is susceptible to being controlled by the application of a signal $r(t)$, through a function $g(t)$, to obtain a response $y(t)$ [36]. The input–output link is a cause-and-effect relationship with the system, so the process to be controlled relates the output to the input.

There are two types of control systems: open-loop systems (know as no automatic) and closed-loop systems (known as automatic). In open-loop systems, control is proportional to the input and is independent of any previous output. In contrast, closed-loop systems relate the control to both the input and output.

Other definitions explain that in open-loop systems, the output is not compared to the reference input. It is economical but is not have precise. In closed-loop systems, the input is compared with a previous state of the output, and the new output is proportional to the difference between the input and output [37].

The automatic control system used in the prototype uses a PID controller. It is the combination of three types of controllers: proportional, integral, and derivative. The PID controller has its output proportional to the error, added to an equivalent quantity of integral of error, and a proportionate amount of the derived error [38–41]. It is expressed as:

$$G(t) = K_p e(t) + \frac{K_p}{T_i} \int e(t)dt + K_p T_d \frac{de(t)}{dt} \tag{1}$$

In which:

$$\frac{K_p}{T_i} = K_i \tag{2}$$

$$K_p T_d = K_d \tag{3}$$

Using Equations (2) and (3), Equation (1) can be expressed as:

$$G(t) = K_p e(t) + K_i \int e(t)dt + K_d \frac{de(t)}{dt} \tag{4}$$

where $T_i$ is the integration time and $T_d$ is the derivative time, while $e(t)$ is the difference between the input and the output. The Equations (2) to (4) are used to obtain a function for implementing the PID controller into a microcontroller. Additionally, thanks to an example of Microchip for implementing a PID controller for a PIC18 MCU [42]. Equation (4) can be reinterpreted as:

$$PID_{value} = PID_p + PID_i + PID_d \tag{5}$$

Proportional part can be interpreted as:

$$PID_p = (K_p)(PID_{error}) \tag{6}$$

When interpreting the integral part, next equation is obtained:

$$PID_i = PID_{i.previous} + (K_i)(PID_{error}) \tag{7}$$

Additionally, the result of the derived is:

$$PID_d = Kd\frac{PID_{error} - PID_{error.previous}}{T_{sampling}} \tag{8}$$

The difference of time between measuring can be interpreted as the sampling time ($T_{sampling}$). Due to the PID controller being used to control the temperature of the heating element, the controller error is:

$$PID_{error} = Temp_{SetPoint} - Temp_{actual} \tag{9}$$

Table 5 summaries what each variable represents.

**Table 5.** Variables and constants for the PID temperature controller.

| Variable | Representation |
|---|---|
| $PID_{value}$ | Value resulting from the PID controller |
| $PID_p$ | Proportional value of the controller |
| $PID_i$ | Integral value of the controller |
| $PID_{i.previous}$ | Last value of the integral part |
| $PID_d$ | Derivative value of the controller |
| $PID_{error}$ | Actual value of the error. It is the difference between the desired value and the current value of the temperature |
| $PID_{error.previous}$ | Last value of the error |
| $Kp$ | Proportional constant |
| $Ki$ | Integral constant |
| $Kd$ | Derivative constant |

## 3. Materials and Methods

The prototype is divided into two principal parts: electronic and structure. The electronic part must perform control tasks, such as temperature control, and offer a control interface for the user. The structure part needs to provide and satisfy different characteristics to improve how users can execute the cutting task. This section introduces the information about the required features and functionality of the prototype and the methodology used for the development of the prototype, proposes the analysis method for a case study, and establishes an approach for a sustainable glass recycling culture proposal.

### 3.1. Methodology

According to Sections 1 and 2, the prototype is developed for its use for individuals and small organizations. Glass recycling grants more benefits for industries with the glass-making process. Most individuals in Mexico skip glass recycling due to reduced monetary gains consequence of the low prices in local markets for glass waste and scrap. By upcycling glass bottles, the new products obtain a new value. The new objects are used for personal use or its sale in local stores. In this way, individuals receive more benefits when upcycling glass bottles, while glass waste reduces and the environment improves.

The prototype requires a temperature variable to control the behavior of the heating element during the process. The PID controller is proposed as the best option to keep the heating element at the same temperature level. For implementing the PID, Equations (6)–(8) are required for controlling the temperature. One of the proposed ideas is to vary the Direct Current (DC) motor velocity to verify if it can improve the cutting process. The glass bottle touches the heating element once. Rotating the bottle is essential to keep the

same temperature on the cut line. This controller is open-loop, and no variable is required as feedback because a Pulse Width Modulation (PWM) signal is used. Thus, the unique variable used for controlling is the temperature. A thermocouple type-k and the MAX6675 module are proposed for this task—see Figure 3.

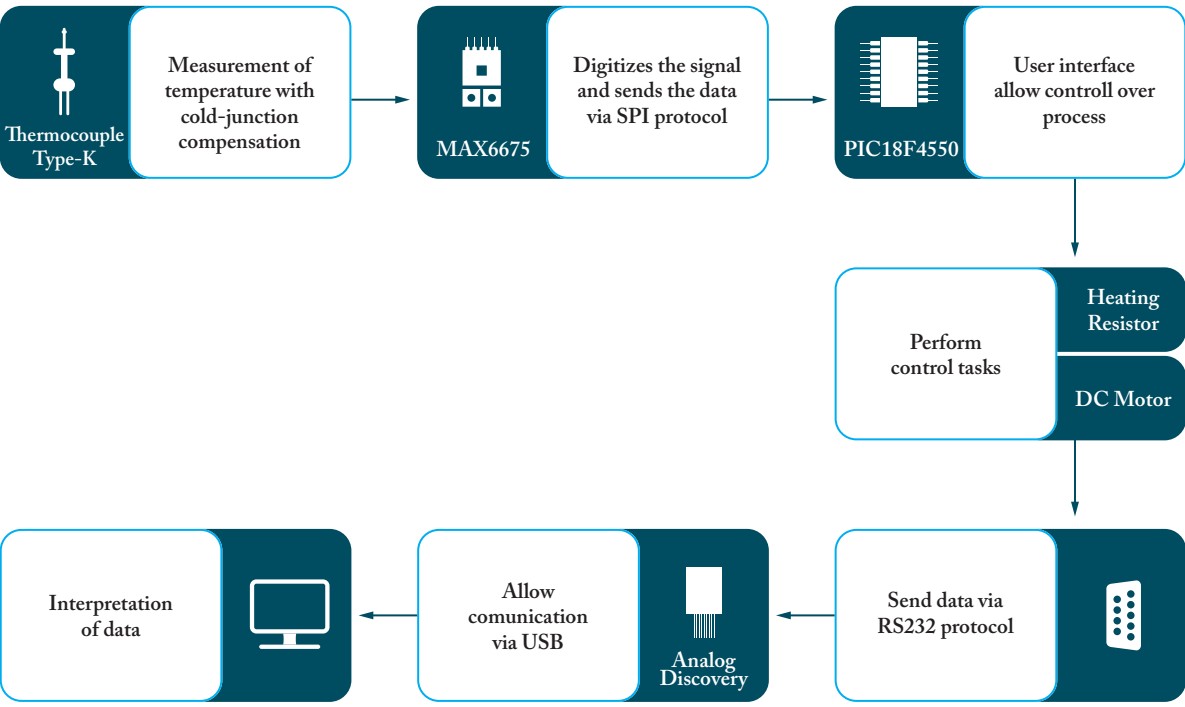

**Figure 3.** Proposed general process diagram.

The reason for both controllers is to reduce the efforts of users. It aims to redirect these efforts to perform the cutting process more times. Consequently, the number of upcycled bottles is increased. The user can change the value desired for both controllers.

The following steps allow us to design the prototype structure:

1. Collecting various and different commercial glass bottles available in the area of study.
2. Using Figure 2 as reference, we acquired the values and data of each bottle:
   - Neck diameter: The minimum value of the neck diameter in the bottle. It is recommended to obtain the value immediately down the ring. The neck diameter is used to establish the values for the glass bottle holder. The value for entry is the principal to determine.
   - Finish height: The value is the difference between the ring and the entrance. It permits the determination of the value of the height of the space for the glass bottle holder.
   - Body diameter: The maximum value of the body diameter is required if the shape is not uniform. It allows us to determine the range for adapting the position of the heating element.
   - Bottle height: The value is the difference between the entrance and the base. It lets us establish the maximum height value of the bottle capable of getting cut.
   - Color type: The data about the color of the glass are not determinant for prototype design.
3. Develop the first design for the prototype structure.
4. Use the data obtained from point 2 to establish dimensions of the structure.
5. Consider some requirements for a good performance of the structure of the prototype.
6. In case the design of the prototype did not consider requirements or can be improved to reach a higher level of accomplishment, the design should be changed or improved.
7. Make sure data are obtained by computer for their interpretation.

8.    Evaluate if the proposed process operates according to expectations.

Glass bottle upcycling generates little glass waste. The proposed solution for a sustainable glass recycling culture considers an ISWM due to the dimensions and principles involved. The following steps are used for the ISWM approach.

1.    Using the Faculty of Engineering as the base for the ISWM proposal, analyze all elements (process) for sustainable culture considering glass waste reduction by recycling and upcycling.
2.    Explain each element, the difficulties involved, the benefits obtained, and the areas needing improvement.
3.    When the elements are identified, all stakeholders must be detected in their element correspondents. Each element must have at least one stakeholder. One stakeholder can participate in one or more elements.
4.    Signalize the responsibilities that each stakeholder has in the element.
5.    Identify all aspects of the ISWM and indicate which element is used.
6.    Explain the importance of each aspect and how it affects the ISWM.

By analyzing the ISWM, realized when sustainability is the objective, and the kind of stakeholders involved, a relation with the Quintuple Helix Model was highlighted. So, the last step for the methodology is:

1.    Gather the information about elements, stakeholders, and aspects from ISWM.
2.    Elaborate on the Quintuple Helix Model using the ISWM approach as a base.

### 3.2. Electronic Circuit

Table 4 exposes two factors for checking and controlling, temperature cycle and material thickens. The thickness of the material varies depending on the brand. A group of bottles of the same kind usually have the same thickness, but different bottle brands may have different thicknesses. The temperature cycle may vary according to the initial temperature of the bottle and the temperature present in the heating element. The difference between both temperatures is known as the temperature gradient. Controlling temperature gradient affects the time needed for cutting. It can be known as a time gradient. The time gradient should be smaller if the temperature difference is significant. One of the ideas for the prototype is to have control over the rotation velocity of the bottle during the cutting process. It aims to determine if rotation velocity can affect the time to perform the cuts.

The electronic circuit aims to control the cutting process. It is necessary to accomplish the following principal points to achieve this:

•    Obtaining and controlling the temperature present over a heating resistor.
•    Varying the velocity present in a DC motor.
•    Sending data to a computer for their interpretation.

Figure 3 displays the proposed general process diagram that the prototype follows.

Although the heating element forms part of the electronic circuit, it is not placed in the same way. It is placed apart from the structure, which considers materials to avoid ignition due to high temperature. Table 6 presents characteristics of main components for the circuit.

In the case of Analog Discovery, an portable oscilloscope designed by Analog Devices and XILINX allows one to measure, visualize, analyze, record, and control mixed-signal circuits. This device is powered by a high-speed Universal Serial Bus (USB) port and the free WaveForms software.

**Table 6.** Characteristics of main components of electronic circuit.

| Characteristic | Details |
| --- | --- |
| **PIC18F4550** | |
| 10-Bit Analog-to-Digital Module | 13 |
| Capture/Compare/PWM Modules | 1 |
| Serial Communications | MSSP, Enhanced USART |
| Universal Serial Bus Module | 1 |
| Comparators | 2 |
| **MAX6675 Integrated Circuit** | |
| Data Output | 12-bit |
| Resolution | 0.25 °C |
| Thermocouple Accuracy | 8 LSBs |
| Maximun Temperature Reading | +1024 °C |
| Digital Conversion | Direct digital conversion of Type-K Thermocouple Output |
| Signal Compensation | Cold-Junction Compensation |
| **Thermocouple Type-K** | |
| Temperature Range | −200 °C to +1250 °C |

### 3.3. Structure

Three parts conform to the structure of the prototype. Each one requires specific characteristics. The three parts are:

- **Support structure**: this is the principal structure in charge of keeping the prototype stable.
- **Heating resistor holder**: it has to hold the heating elements to perform the cut over the bottle.
- **Glass bottle holder**: the most important part, it needs to keep stable the glass bottle during the cutting process.

The structure of the prototype needs to be comfortable and straightforward for its use. If it is hard to use, then the time to accomplish the process will be longer. Table 7 contains the requirements and considerations for each part.

**Table 7.** Requirements and considerations for structure design.

| Requirements | Considerations |
| --- | --- |
| **Support structure** | |
| The structure must be stable during the cutting process | The structure will need to consider a safe way to move from one place to another when required |
| The design should be as simple as possible | It will consider a way to resolve temperature problems present in the structure to avoid making injuries to the user |
| Adapting the height for cutting the bottle must be fast | |
| **Heating element holder** | |
| The position should be stable during the process to avoid variations in cuts | The holding arms need to consider a way to prevent current to flow through it, in case of an accident. |
| Adjusting to the bottle body or neck width must be fast | |
| Due to the heating element being placed here, the holder must consider a way to prevent ignition | |
| **Glass bottle holder** | |
| The grabbing of the bottle should be stable during the cutting process to avoid the bottles falling | Placing the bottle in the prototype needs to be easy to minimize the effort of the user and saving time |
| The holder needs to adapt to the neck diameter of the bottle | |

## 4. Results

In this section, data and the results obtained are presented. Results are introduced in three subsections. These allow us to view the behavior of the prototype and analyze its functionality. Section 4.1 details the prototype design, analysis, and its characteristics. The ISWM and the Quintuple Helix Model proposals are introduced in Section 4.2. It explains both models and details their components. Section 4.3 analyzes new opportunities according with those presented in Sections 4.1 and 4.2.

### 4.1. Structure of Prototype: Design and Performance

Analyzing different glass bottles permitted designing some parts of the structure and establishing its dimensions. Table 8 contains the results of this analysis focused on specific aspects of the industrial glass bottle.

**Table 8.** Measurements of some characteristics of different glass bottles parts.

| Mark of Bottle | Neck Diameter | Helix Height | Body Diameter | Bottle Height | Color Type |
|---|---|---|---|---|---|
| *Café Olé*® 281 mL | 41 mm | 19 mm | 65 mm | 152 mm | Crystalline |
| *Snapple*® 473 mL | 35 mm | 13 mm | 72 mm | 175 mm | Crystalline |
| *Sprite*® 235 mL | 25 mm | 18 mm | 56 mm | 189 mm | Green |
| *BÚHO*® 355 mL | 24 mm | 10 mm | 61 mm | 230 mm | Crystalline |
| *Source Perrier*® 330 mL | 24 mm | 19 mm | 68 mm | 190 mm | Green |
| *del Valle*® 413 ml | 35 mm | 12 mm | 64 mm | 182 mm | Crystalline |
| *Jumex*® 450 mL | 34 mm | 17 mm | 68 mm | 193 mm | Crystalline |
| *Chela Libre*® 660 ml | 25 mm | 17 mm | 75 mm | 287 mm | Amber/Brown |
| *Riunite Lambrusco*® 750 mL | 27 mm | 16 mm | 86 mm | 296 mm | Amber/Brown |
| *Caribe Cooler*® 300mL | 25 mm | 19 mm | 60 mm | 195 mm | Crystal |

Figure 4 presents the designed and implemented structures.

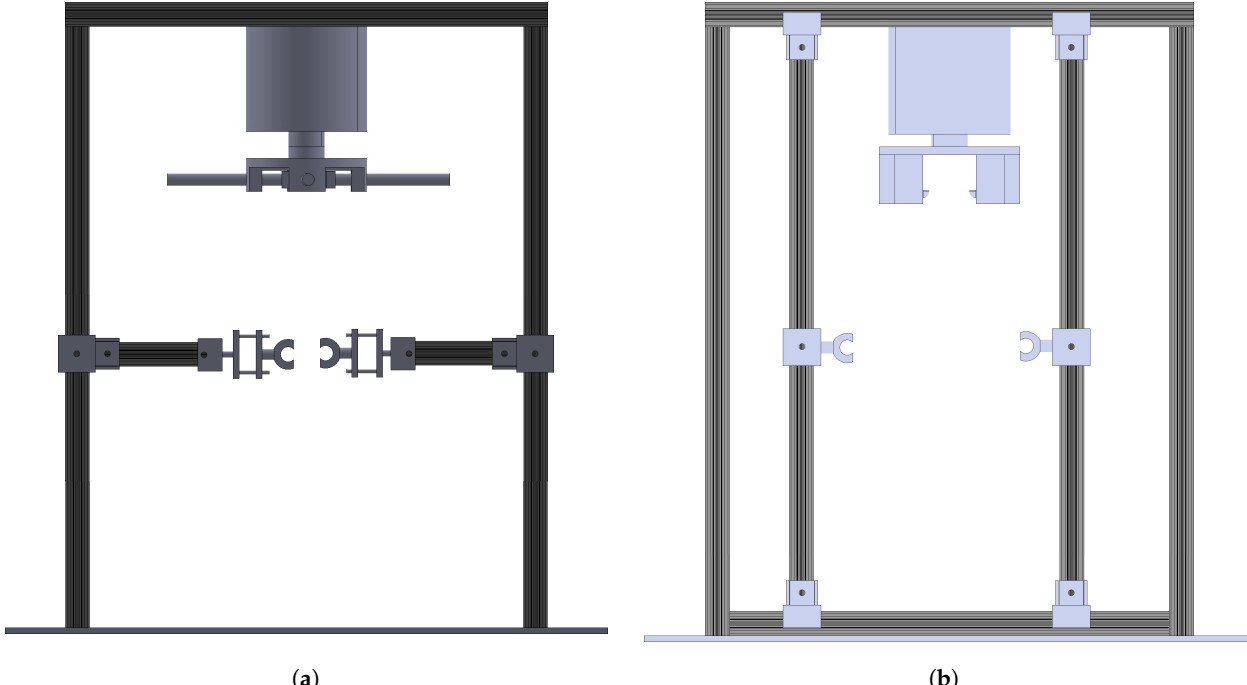

      (**a**)       (**b**)

**Figure 4.** Front view of the prototype design. (**a**) First design; (**b**) last design.

Figure 4a represents the first design implemented. According to the step 5 of the methodology, the next improvements were found:

- The grab of the glass bottle holder is not stable enough and was difficult to use.
- Despite the holder adapting correctly to the neck diameter, its design may be improved.
- Adjusting the heating element took more time than expected.
- The support structure is not as stable as desired.

In accordance with step 6 of the methodology, the prototype design was improved. Figure 4b presents the new design obtained. All the improvements performed over the first design were:

- The movement of heating element holders now is horizontal and requires vertical support. The previous design considered a horizontal axis for its movement. It was deformed due to the weight of the heating element and its holders.
- The new design considers a faster and smooth horizontal movement.
- The glass bottle holder is now designed for its adaptation with only the neck width. The new design is more compact.
- The grab of the bottle, while it is rotating, is more stable.
- The heating element holder uses Bosch aluminum profile and polytetrafluoroethylene (PTFE) bar. Due to their characteristics, these materials are not ignited when high temperatures are reached. The PTFE also avoids electrical current flowing to the prototype structure.

The new design meets all considerations from Table 7. Due to the generated heat mainly affecting the heating holders, the support structure remained at a lower temperature. It allowed us to use this part to move it. Additionally, it resolves problems to avoid injuries by high temperatures when adapting to the width. The heating element holder considers a material to prevent the current from flowing through the prototype structure in case of an accident. This material withstands high temperatures. Figure 5 displays the developed prototype.

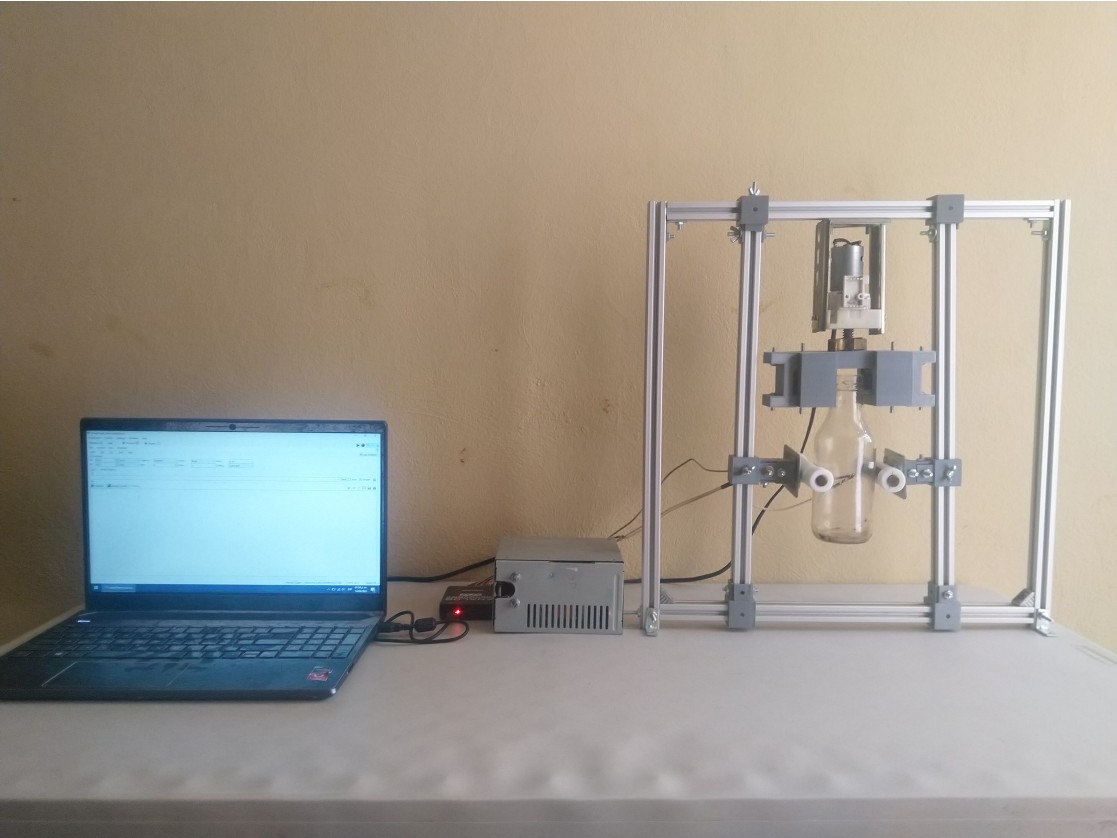

**Figure 5.** Workbench—prototype developed.

When the heating element is at maximum capacity, the average consumption current is 5.3 A. Considering the nominal voltage in Mexico (110 V RMS), the maximum power consumption is 583 W, representing 2.42 W/s. Each cut requires an average power-on period of 15 s. However, the maximum power is not required all time. Figure 6 shows some cuts carried out by using the prototype developed and an example of new products with new use.

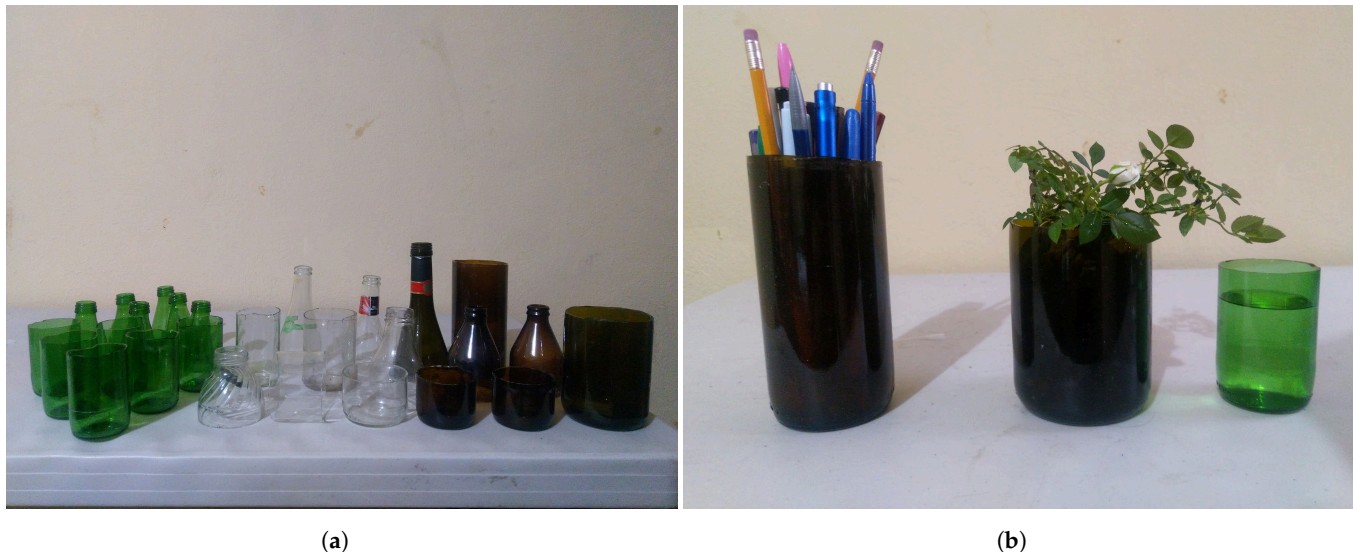

(**a**)                                                                                           (**b**)

**Figure 6.** Glass bottle upcycling: (**a**) example of cuts carried out on glass; (**b**) example of upcycled products.

### 4.2. ISWM and the Quintuple Helix Model: Proposal and Analysis

Certain elements involved in the ISWM are found in the integral glass recycling process, see Figure 1. The use of this project allows the creation of a new branch for it. Figure 7 readapts the integral glass recycling, incorporating the new branch.

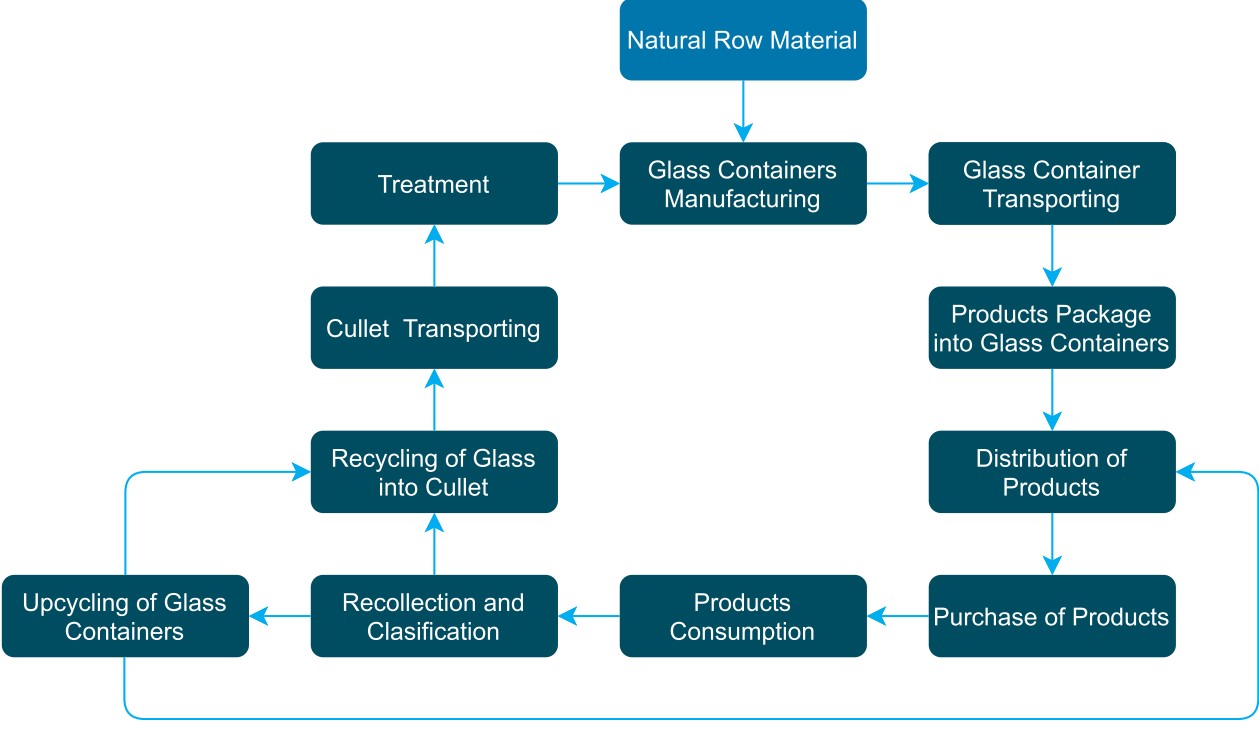

**Figure 7.** A proposed new integral glass recycling process.

The benefits from this proposed process are:

- The objects obtained from glass container upcycling acquire a new value. This means the extension of their life cycle and the creation of new jobs.
- The glass leftover from upcycling is reincorporated in the recycling process as material for obtaining cullet. No glass waste must be generated.
- When glass bottles turn into cullet, a volume reduction is produced. Obtaining cullet before transportation saves transport costs.

Figures 8–10 introduce an approach of each ISWM dimension based on Faculty of Engineering case.

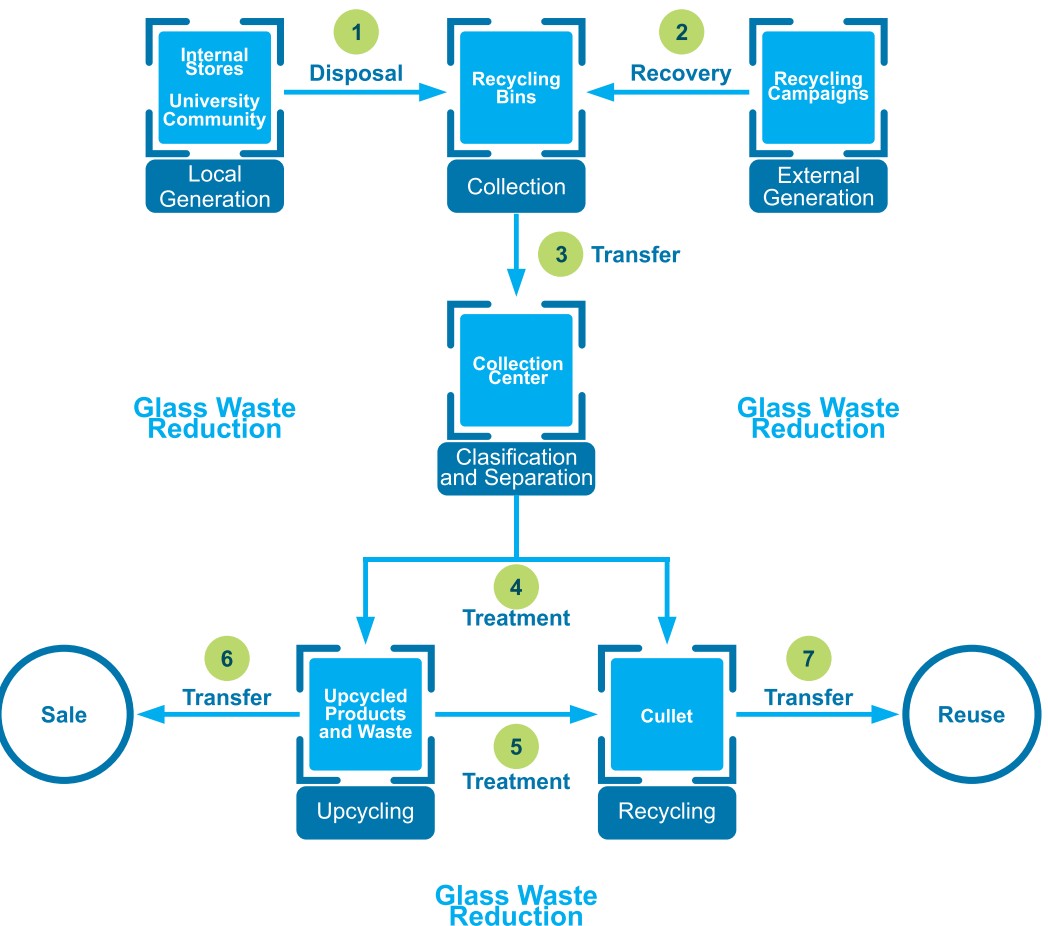

**Figure 8.** Approach for ISWM elements for the Faculty of Engineering case.

Each element displayed represents a process to be executed. These elements are explained in the following points:

- **Local generation and external generation**: These served as input processes and represented the first step to reduce glass waste. Local generation depends on the participation of UAQ users. Inside and around UAQ installations, some stores sell everyday products packed in glass bottles. External generation depends on community participation from Querétaro and requires advertising campaigns to encourage people to participate.
- **Collection**: This element aims to gather all glass waste from local generation and external generation elements. The collection does not mix glass waste.
- **Classification and separation**: The recollection task aims to gather glass containers and keep them in good condition. The classification task helps to verify the state of these and decide which treatment and process are required.

- **Upcycling and recycling**: The upcycling process requires this project for its execution. Glass bottles in a good state are recommended for upcycling. All waste that cannot be used for an upcycled product is used to recycle for cullet generation.
- **Sale and reuse**: Upcycled products represent an opportunity for their sale. The generated cullet represents an opportunity for reuse.

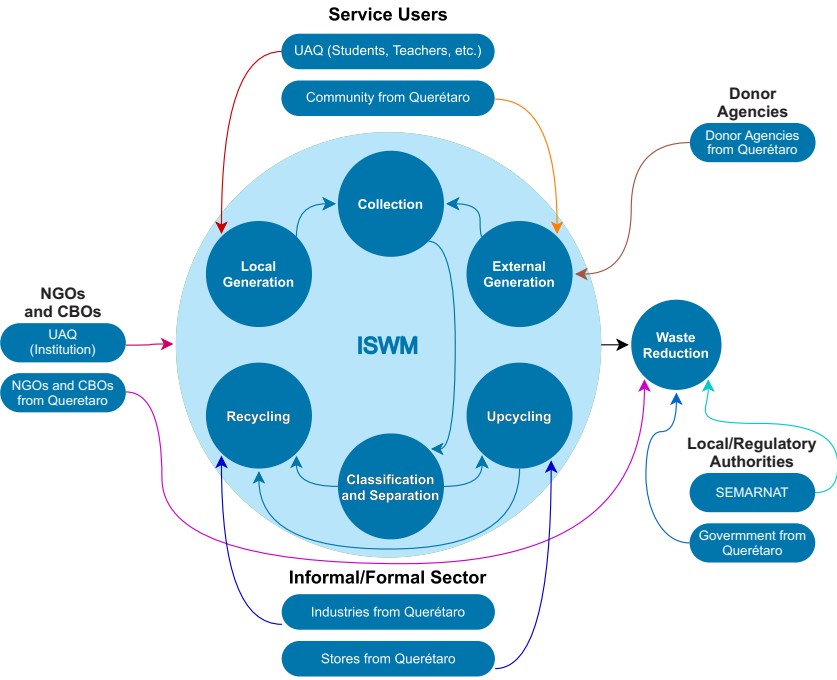

**Figure 9.** Approach for ISWM stakeholders for the Faculty of Engineering case.

All elements need an activity to reach the next element. These activities are required to perform the correct implementation of elements. The following points detail these activities:

- **Disposal**: In FI installations, the containers for this task also have other waste. Due to their limited size, when trash recollection bins are full, students use recycling bins to deposit all kinds of waste. It carries to a bad collection process. Effective promotion, provision, and availability of recycling bins and their design (lids, insert lots, shape, and size) can increase participation and success [43]. It represents an improvement area. If glass bottles are damaged when disposed, they are no longer used for upcycling.
- **Recovery**: External generation origins are recycling campaigns, and some containers are required to dispose of glass waste obtained. It is recommended to have recycling bins that consider classifying the glass waste by color.
- **Transfer (3)**: This activity requires not damaging the glass bottles for upcycling and not mixing the glass waste classified by color.
- **Treatment (4)**: It aims to prepare the glass bottles for upcycling or recycling. If the glass bottle surface is in a perfect state, but with impurities inside or over it, a treatment process must be carried out to guarantee a good cutting process. Different treatments must be carried out to ensure high-quality cullet if the glass bottle or waste is not in a perfect state or presents impurities.
- **Treatment (5)**: It is the same case as Treatment (4). The waste produced by upcycling requires treatment in case new impurities appear.
- **Transfer (6)**: Due to upcycled products obtaining new values, and it is required to keep them in good condition to succeed in their sale.
- **Transfer (7)**: The cullet obtained must conserve its quality during its transfer. If new contaminants mix with the cullet, it affects the process that uses this cullet.

Some stakeholders (UAQ institution and its users, community from Querétaro, industries, and stores) are already identified in the last paragraphs. However, more stakeholders take advantage and participate more in the ISWM. Figure 9 displays the relation between stakeholders with elements.

Different stakeholders can participate in one or more elements depending on the context, diffusion, development, and development of the proposed activity. Industry can participate as a stakeholder for a recycling campaign (external generation).

Some aspects need to be considered for the ISWM. Each aspect contributes to the development and success. Figure 10 indicates some aspects for the ISWM.

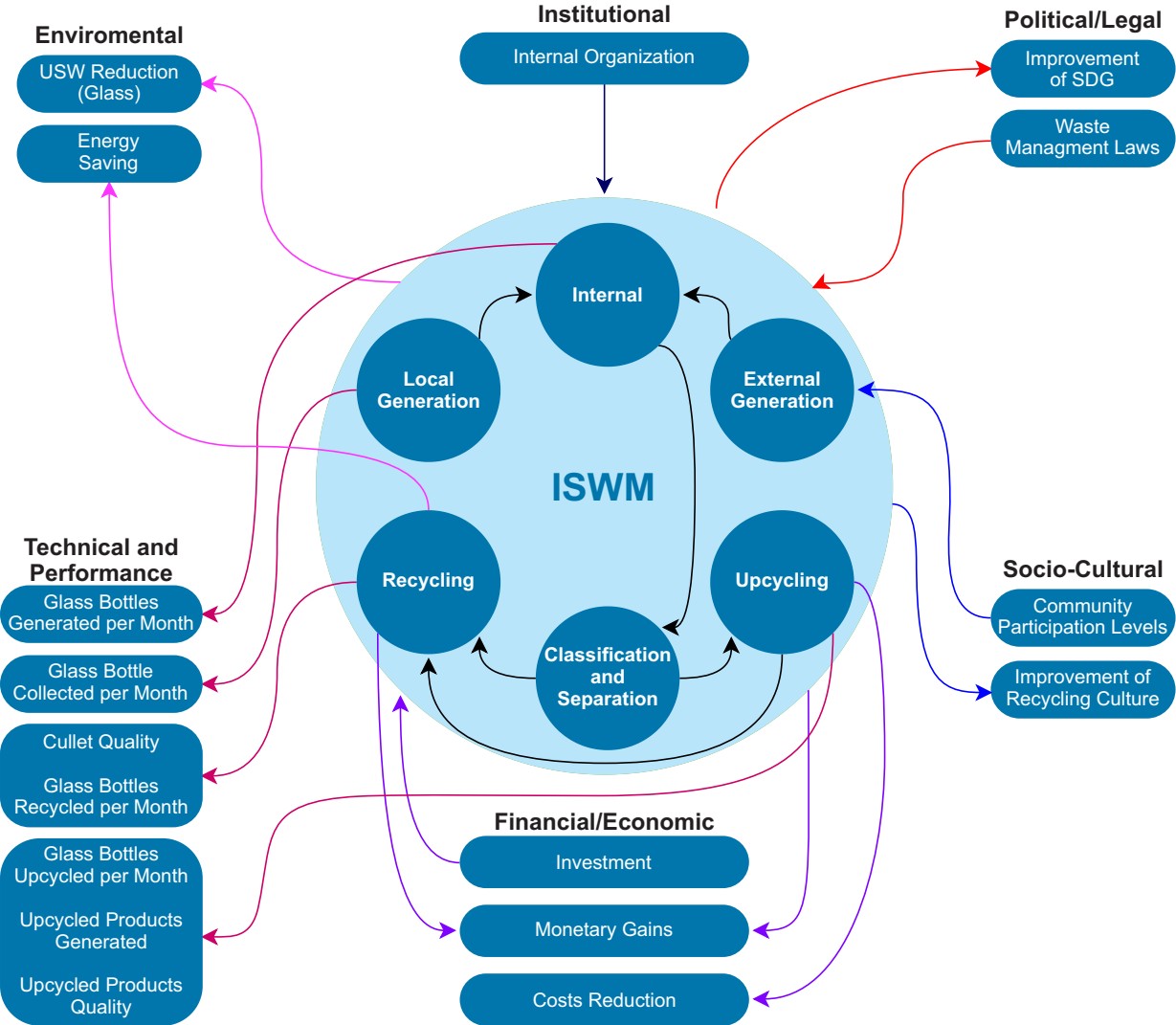

**Figure 10.** Approach for ISWM Aspects for the Faculty of Engineering case.

As a result of using sustainability as an objective to reach, and the relation of it between the ISWM with the Quintuple Helix Model, Figure 11 displays the Quintuple Helix Model using the ISWM proposed.

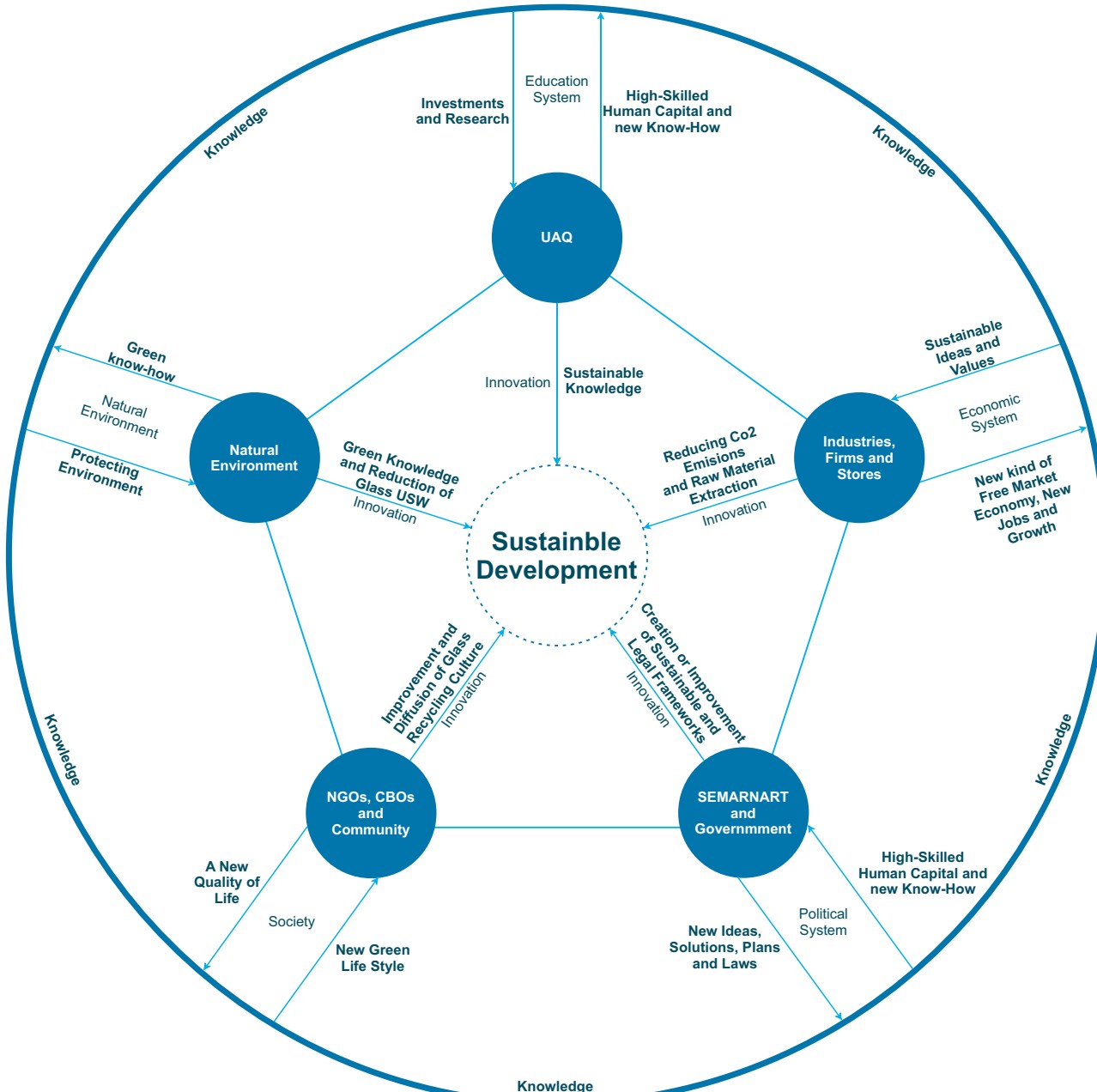

**Figure 11.** Approach for Quintuple Helix Model based on the proposed glass ISWM.

### 4.3. New Opportunities

Additionally, this project presents the opportunity to reach some targets from the twelfth of SDG, which is not considered in Mexico. Table 9 collects the targets and its indicators from SDG 12.

**Table 9.** Opportunities, targets and indicators from goal 12 of SDGs [44].

| | Target | | Indicators |
|---|---|---|---|
| 12.2 | Achieve sustainable management and efficient use of natural resources. | 12.2.1 | Material footprint, material footprint per capita, and material footprint per Gross Domestic Product (GDP). |
| | | 12.2.2 | Domestic material consumption, domestic material consumption per capita, and domestic material consumption per GDP. |
| 12.5 | Substantially reduce waste generation through prevention, reduction, recycling, and reuse. | 12.5.1 | National recycling rate, tons of material recycled. |
| 12.7 | Promote public procurement practices that are sustainable, in accordance with national policies and priorities. | 12.7.1 | Number of countries implementing sustainable public procurement policies and action plans. |
| 12.8 | Ensure that people everywhere have the relevant information and awareness for sustainable development and lifestyles in harmony with nature. | 12.8.1 | Extent to which (i) global citizenship education and (ii) education for sustainable development (including climate change education) are mainstreamed in (a) national education policies; (b) curricula; (c) teacher education; and (d) student assessment. |
| 12.a | Support developing countries to strengthen their scientific and technological capacity to move towards more sustainable patterns of consumption and production. | 12.a.1 | Amount of support to developing countries on research and development for sustainable consumption and production and environmentally sound technologies. |
| 12.b | Develop and implement tools to monitor sustainable development impacts for sustainable tourism that creates jobs and promotes local culture and products. | 12.b.1 | Number of sustainable tourism strategies or policies and implemented action plans with agreed monitoring and evaluation tools. |

Target 12.2 is achieved according to the advantages of glass recycling. A recycling campaign using this project help to reach Target 12.5, 12.7, and 12.8. An example of a campaign using this project to achieve these three targets requires the next point:

- Introduce glass recycling and upcycling as an opportunity to reduce USW;
- Exhibit the proposed integral glass recycling process and its benefits, see Figure 7;
- Present this project as an opportunity to reduce glass waste and create jobs;
- Introduce examples of products obtained by upcycling glass bottles;
- Promote sustainable practices for reducing, recycling, and reusing glass bottles;

According to the Agenda 2030 [45], civil society, the private sector, academia, and governments must unite to achieve the SDG in Mexico:

- The society is invited to join various projects, solve problems in the community and disseminate what has been carried out to meet the objectives.
- The private initiative strengthens alliances and certifies and participates in meeting the sustainable development objectives.
- The academy strengthens research collaboratively to create innovative solutions and measure the impact.
- The government improves the use and application of public resources and generates effective alliances in a participatory model to meet the SDGs.

The project represents an opportunity for it. The academy, in this case, is confirmed by *Universidad Autónoma de Querétaro* (UAQ); the private initiative represents the industry who can take advantage of cullet obtaining and cost reduction, the government aims to reduce USW and create jobs, and society is characterized by people who can participate and receive benefits from all these.

Due to the project being focused on improving the natural environment by reducing glass waste, a Quintuple Helix Model appears. The Quintuple Helix Model represents

the collaboration of the education system, the economic system, the political system, the media-based and culture-based public (society), and the natural environment [31]. In this model, new knowledge (input) is continually stimulated by the circulation of know-how (output). Through innovation from each helix, sustainable development is reached. Further suggestions and impulses for knowledge creation are obtained from investments in the education system and also to contribute to new educational skills and methodologies to transfer knowledge [46–48]. Examples of investments are the article about recycled polyolefins [49] and the article about waste management in Australia [50]. An excellent case for implementing this project is in UAQ. Due to Covid-19, UAQ has implemented some recycling campaigns with the objective of fundraising to develop the "QUIVAX" vaccine. In the beginning, some campaigns consider a wide range of materials for their recollection. Still, the new ones only recollect paper, cardboard, plastic caps, and plastic. The glass was one of those materials to be skipped. Additionally, there is a permanent recycling campaign that considers cardboard, pet, and aluminum. An more specific example occurs in the Faculty of Engineering of UAQ. Despite there being some containers to deposit glass waste, there are no recycling campaigns focused on glass.

## 5. Conclusions

The present article highlights the absence of a recycling culture focused on glass in Mexico, despite its multiples benefits. Affected by various factors (waste price, participation levels, community opinion, and lack of divulging), glass recycling is not as popular as similar activities. It induces people the idea that recycling glass is a waste of time, reducing participation levels and keeping low prices in local markets for glass waste. This cycle does not allow the excellent development of the glass recycling culture. The prototype implementation and the proposed ISWM create an opportunity to break this cycle due to the benefits for stakeholders. This permits a positive change to occur in community opinion and participation levels while reducing glass waste. Additionally, this paper demonstrates the relation between an ISWM and the Quintuple Helix Model when sustainability is required. Each stakeholder from the ISWM is identified in one of the five helices of the model. The benefits of implementing the elements of the ISWM serve as outputs and inputs for each helix, and others aspects serve as the innovation of these. This project represents a perfect example of input (investments and research) for the education system. By developing the prototype, new know-how was acquired. The new know-how is knowledge that circulates and creates new inputs for each of the other four helices. As a result, a sustainable culture focused on glass recycling is proposed.

**Author Contributions:** Conceptualization, J.G.G. and J.R.R.; methodology, J.G.G.; software, J.G.G.; validation, J.G.G., J.R.R., H.R.R., J.M.Á.-A., and O.R.A.; formal analysis, J.G.G., J.M.Á.-A., and O.R.A.; investigation, J.G.G., J.M.Á.-A., and O.R.A.; resources J.G.G., J.M.Á.-A., and O.R.A.; data curation, J.G.G.; writing—original draft preparation, J.G.G.; writing—review and editing, J.G.G., J.R.R., H.R.R., J.M.Á.-A., and O.R.A.; visualization, J.G.G. and H.R.R.; supervision, J.R.R.; project administration, J.R.R.; funding acquisition, J.R.R. All authors have read and agreed to the published version of the manuscript.

**Funding:** This research was funded by Universidad Autónoma de Querétaro.

**Institutional Review Board Statement:** Not applicable.

**Informed Consent Statement:** Not applicable.

**Data Availability Statement:** No new data were created or analyzed in this study. Data sharing is not applicable to this article.

**Acknowledgments:** Thanks to Universidad Autónoma de Querétaro by contributing with monetary financing.

**Conflicts of Interest:** The authors declare no conflict of interest.

## Appendix A

*Appendix A.1. USW Tables*

**Table A1.** Recollected USW composition from 2012 and 2017.

| Type of USW | 2012 | | 2017 | |
|---|---|---|---|---|
| | Percentage | kt | Percentage | kt |
| Organic waste | 52.4 | 22,060 | 51.6 | 23,043 |
| Paper, cardboard and paper products | 13.8 | 5810 | 14.2 | 6338 |
| Others [1] | 12.1 | 5094 | 12 | 5350 |
| Plastics | 10.9 | 4589 | 11 | 4929 |
| Glass | 5.9 | 2484 | 6.1 | 2718 |
| Aluminum | 1.7 | 716 | 1.8 | 798 |
| Textile | 1.4 | 589 | 1.4 | 642 |
| Ferrous metals | 1.1 | 463 | 1.2 | 522 |
| Other non-ferrous metals | 0.6 | 253 | 0.7 | 307 |
| Total | 100 | 42,058 | 100 | 44,647 |

[1] The category Others includes fine waste and disposable diapers, among others.

**Table A2.** Valuable USW composition from 2012 and 2016.

| Type of USW | 2012 | | 2016 | |
|---|---|---|---|---|
| | Percentage | kg/day | Percentage | kg/day |
| Paper, cardboard and paper products | 32 | 143,187 | 28.3 | 21,746 |
| Others [1] | 16.6 | 74,364 | 4.3 | 3274 |
| PET | 15.8 | 70,798 | 22.2 | 17,044 |
| Glass | 13.8 | 62,052 | 18.5 | 14,188 |
| Plastic | 9.2 | 41,115 | 10.3 | 7880 |
| Electronics and domestic appliances | 5.1 | 22,842 | 2.3 | 1788 |
| Iron, sheet and steel | 4.9 | 21,868 | 10 | 7652 |
| Aluminum | 1.4 | 6129 | 2.5 | 1950 |
| Copper, bronze and lead | 1.2 | 5710 | 1.8 | 1346 |
| Total | 100 | 448,065 | 100 | 76,868 |

[1] The category Others includes fine waste and disposable diapers, among others.

**Table A3.** Percentage of USW recycled in various countries according to OECD.

| Country | 2012 | 2014 | 2016 | 2018 |
|---|---|---|---|---|
| Australia | 32 | 30 | 28 | - |
| Austria | 25 | 26 | 26 | 26 |
| Belgium | 33 | 33 | 33 | 34 |
| Czech Republic | 21 | 23 | 27 | 23 |
| Denmark | 27 | 28 | 30 | 32 |
| Estonia | 18 | 29 | 28 | 26 |
| Finland | 22 | 18 | 29 | 29 |
| France | 21 | 22 | 23 | 25 |
| Germany | 46 | 48 | 49 | 49 |
| Greece | 14 | 12 | 14 | 15 |
| Hungary | 21 | 25 | 27 | 29 |
| Iceland | 24 | 21 | 26 | - |
| Ireland | 34 | 34 | 34 | 30 |
| Italy | 25 | 27 | 30 | 32 |
| Japan | 20 | 21 | 20 | 20 |
| Korea | 58 | 58 | 59 | 64 |
| Lithuania | 20 | 21 | 26 | 27 |
| Luxembourg | 28 | 28 | 29 | 30 |
| Mexico | - | - | - | - |
| The Netherlands | 24 | 24 | 26 | 27 |

**Table A3.** *Cont.*

| Country | 2012 | 2014 | 2016 | 2018 |
|---|---|---|---|---|
| Norway | 26 | 26 | 28 | 31 |
| Poland | 13 | 21 | 28 | 26 |
| Portugal | 12 | 16 | 15 | 13 |
| Slovak Republic | 8 | 5 | 16 | 17 |
| Slovenia | 38 | 30 | 48 | 54 |
| Spain | 20 | 17 | 18 | 18 |
| Sweden | 32 | 33 | 32 | 30 |
| Switzerland | 35 | 33 | 31 | 31 |
| Turkey | - | - | 9 | 12 |
| United Kingdom | 27 | 27 | 27 | 27 |
| United States | 26 | 26 | 26 | 25 |

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
