# Peer review of "Sustainable Glass Recycling Culture-Based on Semi-Automatic Glass Bottle Cutter Prototype"

_sustainability, doi:10.3390/su13116405_

Round 1
Reviewer 1 Report
- In Table 1: As glass is very stable, it is confusing how recycling glass will reduce air pollution, water pollution and CO2 emission?
- Table 6 is taken a lot of space while it is useless to the readers as it can be briefly described using a few sentences.
- In the introduction section, the authors took a large space to explain the low rate and difficulties of recycling glass. However, it is not clear how the methodology proposed by this research will help to address the difficulties to recycle glass.
- In Section 2, the authors list several subsections and give some descriptions for each subsection. However, it can hardly see any connections between them.
- I am wondering the Prototype developed can only cut the glass bottle into very big pieces, how it can contribute to cut into small pieces- cullet? Also, it seems very time-consuming to cut the glass one by one, I don’t think it has practical meaning.
- Are there any photos that the authors can show the cutting result from the Prototype developed?
Author Response
Authors appreciate enormously your kind feedback. We hope these changes fulfill your expectations.
In Table 1: As glass is very stable, it is confusing how recycling glass will reduce air pollution, water pollution and CO2 emission?
Thank you for the observation. We deleted Table 1 for better comprehension. Its content is shown now in lines 41-69. Glass recycling is a great activity for sustainability, promoting a considerable reduction in air and water pollution of 20 and 40-50% in the supply chain. We added to the manuscript this information in lines 62-67.
Environmental contaminants decrements: Glass recycling is one technique that positively impacts sustainability since it is possible to reuse this raw material several times. Turning it into an alternative raw material by not losing its quality makes it a high proposal [13]. According to [14,15], when a glass tone is recycled, 0.58 t of CO2 is saved, air pollution is reduced by 20%, and water pollution 40 to 50% throughout the supply chain.
Table 6 is taken a lot of space while it is useless to the readers as it can be briefly described using a few sentences.
Thank you very much for the feedback. Table 6 was removed from the manuscript. We present that information in lines 119-121.
In the introduction section, the authors took a large space to explain the low rate and difficulties of recycling glass. However, it is not clear how the methodology proposed by this research will help to address the difficulties to recycle glass.
Thank you very much; we appreciate your comment. We have restructured Section 1 for better comprehension. The actual Subsection 1.3 explains in lines 126-141 the aim, the target users, and how the proposed project helps solve glass recycling problems. Also, Section 3 is now presented as Materials and Methods. Subsection 3.3 is updated by adding lines 301-308. It explains the methodology used for developing the solution proposed in lines 326-369.
In Section 2, the authors list several subsections and give some descriptions for each subsection. However, it can hardly see any connections between them.
Thank you very much for the observation. As aforementioned, certain sections and tables were deleted and better described. It can be observed in lines 190-192 for the bottle. Also for thermal shock in lines 209-211. Finally for the electronic control the lines 224-227.
Thank you very much for the observation. Section 2 is updated. The previous Subsection 2.3 and 2.4 are combined in actual Subsection 2.5 (Temperature Control). The paragraph in lines 148-152 introduces all five Subsections. Also, we present now the frameworks about ISWM and Quintuple Helix Model here. The subsections are presented in sequence, taking Integrated Sustainable Waste Management of our proposal work as a starting point: sentences are added to connect them (lines 165-167 for the Quintuple Helix Model, lines 196-198 for glass bottles, lines 215-217 for Thermal Shock, and lines 229-232 for Temperature Control) for better comprehension.
I am wondering the Prototype developed can only cut the glass bottle into very big pieces, how it can contribute to cut into small pieces- cullet? Also, it seems very time-consuming to cut the glass one by one, I don’t think it has practical meaning.
We appreciate your comment. Recycled glass is generally targeted for industrial use due to the low cost of mass production and subsequent benefits. Our proposal promotes upcycling activities, allowing creating pieces by making strategic cuts to recover and reuse parts of bottles that can be used as art, crafts, and objects with more value. This is also why the time increases because the cuts are made at different previously identified points to obtain various pieces. The glass waste generated by upcycling is used for cullet generation. It aims to zero glass waste production.
Are there any photos that the authors can show the cutting result from the Prototype developed?
We appreciate your comment. Figure 6 shows some cuts done on glass bottles by using the prototype developed.
Reviewer 2 Report
In this work sustainable glass recycling approach based on semi-automatic glass bottle cutter prototype has been described. It is hoped that by implementing new recycling technologies, recycling of glass wastes will increase, especially in countries like Mexico where still most of glass wastes is not recycled. According to Integrated Sustainable Waste Management (ISWM) and the Quintuple Helix Model, a sustainable approach toward glass recycling was proposed. It is a valuable method to (i) support the recycling culture in the society, and (ii) help to solve (at least partially) the problems with glass recycling. The results presented are well-structured and logically presented, however, some points need to be addressed in more depth:
- what about the impurities in or on the surface of the bottles? How do they influence on the recycling process and the quality of the recycling products?
- What is the energy consumption of the recycling process? It is important in sustainable economy to know mass and energy balance of the process,
- it is "a semi-automated prototype focused on recycling glass bottles using the thermal shock phenomenon" - please explain the details of the thermal shock - what is the temperature? If it is very high, the prototype must be protected from ignition itself (e.g. in the electronic compartment) and the subsequent fire,
- Concerning the ISWM and the Quintuple Helix Model please show the experimental validation of the models.
- The Introduction contains seven tables - please reduce their number to improve the flow and readability.
Author Response
Authors appreciate enormously your kind feedback. We hope these changes fulfill your expectations.
In this work sustainable glass recycling approach based on semi-automatic glass bottle cutter prototype has been described. It is hoped that by implementing new recycling technologies, recycling of glass wastes will increase, especially in countries like Mexico where still most of glass wastes is not recycled. According to Integrated Sustainable Waste Management (ISWM) and the Quintuple Helix Model, a sustainable approach toward glass recycling was proposed. It is a valuable method to (i) support the recycling culture in the society, and (ii) help to solve (at least partially) the problems with glass recycling. The results presented are well-structured and logically presented, however, some points need to be addressed in more depth:
- what about the impurities in or on the surface of the bottles? How do they influence on the recycling process and the quality of the recycling products?
Thank you very much; we appreciate your comment. If a glass bottle presents impurities, the cut and thermal shock are affected. The cut may not be uniform, and the thermal shock may take more time. A treatment process is recommended before upcycling or recycling. We specify this situation in lines 460-464.
- What is the energy consumption of the recycling process? It is important in sustainable economy to know mass and energy balance of the process,
Thank you very much for your feedback. In lines 414-418, we introduce the energy consumption of the process. When the heating elements are at maximum capacity, an average consumption current is 5.3 A. Considering the nominal voltage in Mexico (110 V RMS), the maximum power consumption is 583 W. Which represents a consumption of 0.16 W/s. Each cut done requires an average power-on period of 15 seconds. However, maximum power is not required at all times.
- it is "a semi-automated prototype focused on recycling glass bottles using the thermal shock phenomenon" - please explain the details of the thermal shock - what is the temperature? If it is very high, the prototype must be protected from ignition itself (e.g. in the electronic compartment) and the subsequent fire,
Thank you very much; we appreciate your comment. The thermal shock phenomenon now is explained in Subsection 2.4 in lines 218-219. Also, in lines, 284-286 is specified that the heating element is not in the same place as the electronic compartment. Furthermore, the used materials were selected strategically for the hard operating conditions.
- Concerning the ISWM and the Quintuple Helix Model please show the experimental validation of the models.
Thank you very much for the observation. The validation for the ISWM is in lines 159-161 and the validation for the Quintuple Helix Model is in lines 179-195.
- The Introduction contains seven tables - please reduce their number to improve the flow and readability.
We appreciate your comment. We remove Table 1. The information now is introduced in lines 42-70. Old Table 2, Table 3, Table 5 were moved to Annex A1. Also, The information of Table 5 is splitted in actual Table 2 and Table A3, both incorporate new data. Table 6 was removed and its information is resumed at lines 119-121. Old Table 7 is now presented in Subsection 2.1
Reviewer 3 Report
Dear Authors,
I find the manuscript topic of interest for the glass recycling sector. I appreciate the comprehensive work done. Please find below, a few suggestions intended for the manuscript improvement:
- Tabel 2 and Tabel 3: It would be useful to add a line with the Total
- Lines 51-52 Please explain why the decrease occurred: 2012 v.s. 2016
- Tabel 4. Please add the reference year
- Lines 58-63 Please provide some recent data regarding the recycling glass rate in Mexico. The dynamic of the market and ISWM view changed, especially in the last 10 years within the transition from the linear towards the circular economy. I suggest adding a column with the recent rates on USW recycled in the countries mentioned.
- I would explain that Mr. Victor M. Toledo is the Secretary of Environment and Natural Resources of Mexico
Best regards,
Author Response
Authors appreciate enormously your kind feedback. We hope these changes fulfill your expectations.
I find the manuscript topic of interest for the glass recycling sector. I appreciate the comprehensive work done. Please find below, a few suggestions intended for the manuscript improvement:
- Tabel 2 and Tabel 3: It would be useful to add a line with the Total
Thank you very much for the observation. A new line with the Total was added for both cases. Also, both are introduced now in Annex.
- Lines 51-52 Please explain why the decrease occurred: 2012 v.s. 2016
Thank you very much for the feedback. Both data are obtained from reports by SEMARNART (the Mexican office for sustainability). No one of these contains the reason behind the decrement of USW collected that have value from 2012 to 2016. We have explored official reports by SEMARNART and similar documents about USW in Mexico but the reason is unknown. Also, the informal recollection is not considered in this work because their number, organization, and coverage are partially known.
- Tabel 4. Please add the reference year
Thank you very much for the support. The reference year has been added in Table 4.
- Lines 58-63 Please provide some recent data regarding the recycling glass rate in Mexico. The dynamic of the market and ISWM view changed, especially in the last 10 years within the transition from the linear towards the circular economy. I suggest adding a column with the recent rates on USW recycled in the countries mentioned.
We appreciate your comment. We updated the previous Table 5 and compared the recycling rates from the years 2012, 2014, 2016, and 2018 in actual Table 2. Also, data about countries from OECD is provided in actual Table A3.
- I would explain that Mr. Victor M. Toledo is the Secretary of Environment and Natural Resources of Mexico
We appreciate your comment. The information about Mr. Victor M. Toledo is updated on line 116. He actually is the former secretary of SEMARNART.
Round 2
Reviewer 1 Report
The authors somehow improved the manuscript after the revision. However, I do encourage the authors to consider more about the practical significance of their future research.
Reviewer 2 Report
Authors have adequately replied to all comments and questions- the revised manuscript can be accepted.